# Evidence for a Novel Antiviral Mechanism of Teleost Fish: Serum-Derived Exosomes Inhibit Virus Replication through Incorporating Mx1 Protein

**DOI:** 10.3390/ijms221910346

**Published:** 2021-09-26

**Authors:** Jian He, Nan-Nan Chen, Zhi-Min Li, Yuan-Yuan Wang, Shao-Ping Weng, Chang-Jun Guo, Jian-Guo He

**Affiliations:** 1State Key Laboratory for Biocontrol, School of Marine Sciences, Sun Yat-sen University, 135 Xingang Road West, Guangzhou 510275, China; hejian35@mail.sysu.edu.cn (J.H.); 15626412077@163.com (N.-N.C.); lizhm6@mail2.sysu.edu.cn (Z.-M.L.); wangyy99@mail2.sysu.edu.cn (Y.-Y.W.); lsshjg@mail.sysu.edu.cn (J.-G.H.); 2Southern Laboratory of Ocean Science and Engineering (Guangdong, Zhuhai), School of Marine Sciences, Sun Yat-sen University, 135 Xingang Road West, Guangzhou 510275, China; 3Guangdong Provincial Key Laboratory of Marine Resources and Coastal Engineering, Sun Yat-sen University, 135 Xingang Road West, Guangzhou 510275, China; 4Guangdong Province Key Laboratory for Aquatic Economic Animals, School of Life Sciences, Sun Yat-sen University, 135 Xingang Road West, Guangzhou 510275, China; lsswsp@mail.sysu.edu.cn

**Keywords:** teleost fish, exosome, immunity, myxovirus resistance 1, Infectious spleen and kidney necrosis virus

## Abstract

Exosomes are associated with cancer progression, pregnancy, cardiovascular diseases, central nervous system-related diseases, immune responses and viral pathogenicity. However, study on the role of exosomes in the immune response of teleost fish, especially antiviral immunity, is limited. Herein, serum-derived exosomes from mandarin fish were used to investigate the antiviral effect on the exosomes of teleost fish. Exosomes isolated from mandarin fish serum by ultra-centrifugation were internalized by mandarin fish fry cells and were able to inhibit Infectious spleen and kidney necrosis virus (ISKNV) infection. To further investigate the underlying mechanisms of exosomes in inhibiting ISKNV infection, the protein composition of serum-derived exosomes was analyzed by mass spectrometry. It was found that myxovirus resistance 1 (Mx1) was incorporated by exosomes. Furthermore, the mandarin fish Mx1 protein was proven to be transferred into the recipient cells though exosomes. Our results showed that the serum-derived exosomes from mandarin fish could inhibit ISKNV replication, which suggested an underlying mechanism of the exosome antivirus in that it incorporates Mx1 protein and delivery into recipient cells. This study provided evidence for the important antiviral role of exosomes in the immune system of teleost fish.

## 1. Introduction

Exosomes are extracellular vesicles of endosomal origin and vary in size, ranging from ~40 to 160 nm in diameter [1]. Many cell types release exosomes into the extracellular environment and are found in biological fluids, such as semen, amniotic fluid, breast milk and blood [2]. Serum-derived exosomes are the most abundant and widely distributed in the body [3]. A wide range of physiological and pathological functions of exosomes were confirmed and attributed to transferring functional cargos [2]. Proteins, metabolites, and nucleic acids delivered by exosomes into recipient cells effectively alter their biological response [4,5].

In mammals, exosomes are involved in multiple biological and pathological processes, such as reproduction and development, cancer progression, cardiovascular diseases, and immune responses, especially in virus infections [6]. Exosomes can improve viral infectivity by spreading viral and cellular components, resulting in virus immune evasion and depressing the immune response [7]. Moreover, exosomes directly exchange interferon (IFN)-stimulated genes, proteins, or mRNA among host cells, which might contribute to the establishment of an anti-viral state in recipient cells [8]. For example, human macrophages shed exosomes that deliver antiviral mediators, including APOBEC3G (apolipoprotein B mRNA editing enzyme, catalytic polypeptide-like 3G), protecting human hepatocytes from Hepatitis B virus infection [9]; miR-1975 was secreted in exosomes and taken up by the neighboring cells to induce IFN expression, suppressing Influenza virus replication [10]. However, study on the role of exosomes in the immune response of teleost fish, especially antiviral immunity, is limited.

Mandarin fish (*Siniperca chuatsi*) is an economically important species that is widely distributed in China; its industry is threatened by different aquatic viruses, especially the Infectious spleen and kidney necrosis virus (ISKNV) [11]. Iridoviruses are large dsDNA viruses that can infect both invertebrates (particularly insects and crustacean), and poikilothermic vertebrates (fish, amphibians, and reptiles), leading to systematic diseases [12]. ISKNV belongs to species of the genus *Megalocytivirus* from the family *Iridoviridae* [13]. In this study, serum-derived exosomes from mandarin fish were used to investigate the antiviral effect for the exosomes of teleost fish and found that the serum-derived exosomes from mandarin fish could inhibit ISKNV replication by incorporating the Mx1 protein into exosomes. This work provides evidence for the important antiviral role of exosomes in teleost fish.

## 2. Results

### 2.1. Exosomes Were Isolated from Mandarin Fish Serum

In the current study, exosomes were isolated from mandarin fish serum by ultracentrifugation (Figure 1A). The purified exosomes were further identified by Western blot analysis using the antibodies of three representative exosome markers [4], lysosomal associated membrane protein 3 (CD63), tumor suppressor gene 101 (TSG101), and heat shock 70 kDa protein-8 (HspA8) (Figure 1B). Transmission electron microscopy images showed that exosomes have a cup-shaped bilayer-enclosed morphology, ranging from 40 to 150 nm in size (Figure 1C). Furthermore, whether exosomes could internalize into mandarin fish fry (MFF-1) cells was investigated. Exosomes were labeled with green-fluorescent dye PKH67 [14]. PKH67-labeled exosomes were incubated with cells for 2 h, and green fluorescence was detected around the nucleus in MFF-1 cells (Figure 1D). These results indicated that serum-derived exosomes from mandarin fish could internalize into recipient cells.

### 2.2. Serum-Derived Exosomes from Mandarin Fish Inhibited ISKNV Infection

The MFF-1 cells were co-cultivated with exosomes (0, 1, 5, or 10 μg/mL) after ISKNV infection to investigate whether exosomes exert their antiviral function against fish virus. Viral levels were measured via qPCR and Western blot analyses at 48 h post infection. The DNA level of the *isknv-mcp* gene was used to represent the level of the viral genome. As shown in Figure 2A, the ISKNV genome equivalent level obviously decreased after incubation with exosomes in a dose-dependent manner. Similar results were also observed in the levels of the ISKNV VP101L protein (Figure 2B), which is an important structural protein of ISKNV [15]. These results suggested that serum-derived exosomes from mandarin fish inhibited ISKNV infection.

### 2.3. Mx1 Protein Was Incorporated into the Serum-Derived Exosomes

The protein composition of serum-derived exosomes was investigated to further reveal the underlying mechanisms of exosomes in inhibiting ISKNV infection. Myxovirus resistance 1 (Mx1), a critical antivirus component of innate immunity induced by type I IFNs [16], was found in the exosomes by mass spectrometry analysis (Figure 3A) and was then identified by Western blot analysis using anti-Mx1 antibody (Figure 3B). The exosomes were treated with trypsin digestion to further verify the localization of the Mx1 protein in the exosomes. As shown in Figure 3C, the Mx1 protein was also detected when the exosomes were treated with trypsin, whereas the Mx1 protein was reduced in the exosomes treated with Trion X-100 before trypsin. These observations indicated that the Mx1 protein was not incorporated on the serum-derived exosomes surface, but into the serum-derived exosomes.

### 2.4. Mx1 Protein Could Be Transferred to the Recipient Cells through the Exosomes

The Mx1-GFP-tag plasmid was constructed and transfected into MFF-1 cells to further confirm whether Mx1 could be incorporated into cellular exosomes. At 24 h post-transfection, exosomes were isolated from the cell culture supernatant and then analyzed by Western blot. The results showed that the Mx1-GFP recombinant protein was detected in the cellular exosomes (Figure 4A), indicating that this protein was packaged into the exosomes. In addition, Mx1-GFP-labeled cellular exosomes could rapidly endocytose into MFF-1 cells and colocalized with markers specific to late endosomes/lysosomes (Figure 4B). These results suggested that the Mx1 protein could be transferred to the recipient cells though the exosomes.

## 3. Discussion

Teleost fish exosomes were identified in rainbow trout (*Oncorhynchus mykiss*), tongue sole (*Cynoglossus semilaevis*) and zebrafish (*Danio rerio*) [17,18,19], and take part in a variety of physiological processes, such as sex maturation and stress response [18,19,20]. However, the roles of teleost fish exosomes in antiviral effects remain limited. In the present study, the exosomes were isolated and purified from mandarin fish serum, and their antivirus role was investigated. Our results showed that the mandarin fish serum-derived exosomes could internalize into MFF-1 cells and inhibit ISKNV replication, indicating the teleost fish exosomes play an important anti-viral role.

The underlying mechanisms of exosomes antiviral are decided by the cargos they take. A recent study suggested that IFN-induced proteins were proven to be incorporated in the exosome to suppress viruses. The exosomes contain the IFN-induced transmembrane protein to inhibit Dengue virus [21]; human angiotensin-converting enzyme 2 (hACE2), which was recently identified as an IFN-stimulated gene, exists on the surface of exosomes and could specifically block the cell entry of SARS-CoV-2 [22]; toll-like receptor-3 activated human cervical epithelial cells release exosomes, which contain multiple IFN-stimulated genes (ISG56 and OAS1), inhibit Human immunodeficiency virus replication [23]. Herein, the protein composition of serum-derived mandarin fish exosomes was analyzed by LC-MS/MS and identified the Mx1 protein incorporated into the exosomes. Moreover, the exogenous fusion Mx1-GFP proteins could also be incorporated into cellular exosomes. This finding further confirmed that the Mx1 proteins were incorporated into exosomes.

Mx1 is recommended as an IFN-induced GTP-binding protein, belonging to the dynamin superfamily of large GTPases that are associated with intracellular membranes and are involved in a wide range of intracellular transport processes [24]. This is a critical component of innate immunity while most Mx genes are induced by type I IFNs in response to viral infection. Mx proteins are against many RNA and DNA viruses [25,26]. A recent study shows that overexpressed *Andrias davidianus* Mx (AdMx) in the Chinese giant salamander’s muscle cells significantly reduced Chinese giant salamander iridovirus (GSIV) replication indicating that the Mx shows potential antiviral properties against iridoviruses [27]. In the present study, the localization of Mx1 protein in exosomes was identified. The results showed that Mx1 was not on the surface of the exosomes but incorporated into the exosomes. These results imply that this exosome carrying Mx1 performs a function inside the MFF-1 cell. Moreover, the Mx1-GFP-labeled cellular exosomes could endocytose into MFF-1 cells, which indicated that the Mx1 protein could be transferred into the recipient cells though the exosomes. These findings suggested that the anti-viral effect of this serum-derived exosome is the direct delivery of the Mx1 into the recipient cells to suppress ISKNV replication.

In conclusion, our study suggested that the serum-derived mandarin fish exosomes could inhibit ISKNV replication, of which the underlying mechanism of the antivirus function might be the incorporation of the Mx1 protein into exosomes and delivery into recipient cells. This work provides evidence for the important antiviral role of exosomes in teleost fish.

## 4. Materials and Methods

### 4.1. Cells, Viruses, and Animals

Mandarin fish fry cells (MFF-1 cells) were established to originate from the mandarin fish fry by Dr. Dong CF [28]. MFF-1 cells were grown and maintained in Dulbecco’s modified Eagle’s medium (DMEM) supplemented with 10% fetal bovine serum (FBS) at 27 °C in 5% CO_2_ [28]. The MFF-1 cell line was highly susceptible to the ISKNV [27]. In our lab, this cell line is used to ISKNV passage and isolate from disease-infected fish [28].

The ISKNV(NC_003494.1) was originally isolated from disease-infected mandarin fish [13]. The ISKNV strain used in this study was separated and obtained from the Nanhai mandarin fish farm in 2017 and stored in our laboratory [29]. The whole genome of this ISNKV strain has no significant difference to ISKNV(NC_003494.1). This ISKNV strain has been passaged in the MFF-1 cell line for eight generations. The virus titer was determined using the 50% tissue culture infective dose (TCID_50_) method in a 96-well culture plate [30]. The MOI was calculated by the number of virions/the number of cells in the plate well. [31]. The number of virions was calculated by the number of ISKNV genome copies using the standard-curve method of absolute quantitative real-time PCR (qPCR). The number of cells was calculated by a cell count instrument before cells seeding on 24-well plates (at a density of 1 × 10^5^/one well); after 24 h of cultivation, the MFF-1 cells covered the entire well and the density in one well of 24-well plates is 3 × 10^5^. The MOI used in this study is MOI = 1.

Mandarin fish samples (±500 g) were obtained from the fish farms in Foshan, Guangdong, China. Ten mandarin fish were used in the experimental work to obtain the mandarin fish serum. All animal experiments were performed in accordance with the regulation for animal experimentation of Guangdong Province, China and were approved by the Ethics Committee of Sun Yat-sen University.

### 4.2. Plasmid Construction and Transient Transfection

Recombinant DNA techniques were performed according to standard procedures. The primers used in this study were forward (pEGFP-N3-*sc*Mx-F primer 5′-CAATGTAGCACCCGCACTGACC-3′) and reverse (pEGFP-N3-*sc*Mx-R primer 5′-ACCTCACGCTCCTCGCTTGTC-3′) primers. The mandarin fish Mx-1 cDNA sequence (GenBank: AY392097.1) from MFF-1 cells was cloned into pEGFP-N3 empty plasmid to generate mandarin fish Mx1-GFP. The transient transfection of recombinant DNA plasmids into MFF-1 cells was performed using FuGENER HD Transfection Reagent ^®^ (Promega, Madison, WI, USA) according to the instructions of the manufacturer.

### 4.3. Exosome Isolation from Fish Serum and Electron Microscopy Analysis

Fresh blood was collected from the caudal vein of healthy mandarin fish in a sterile tube, incubated at 37 °C for 1 h and then at 4 °C overnight. Then, the samples were centrifugated at 5000× *g* rpm for 15 min at 4 °C and collected the supernatant serum. The serum was diluted with phosphate buffer solution (PBS) at a ratio of 1:10 before centrifugate at 2000× *g* for 30 min at 4 °C. Next, the supernatant was centrifugated at 12,000× *g* for 45 min at 4 °C. The pre-cleared supernatant was transferred to a SW40 pipe and ultracentrifugation at 110,000× *g* for 2 h at 4 °C (SW40, Beckman-Coulter, Inc., Brea, CA, USA). Then the precipitate was resuspended in a large volume of PBS and passed through a 0.22 μm pore PES filter (Merck Millipore, Billerica, MA, USA). This supernatant (pre-cleared medium) was ultracentrifugated at 110,000× *g* for 70 min at 4 °C (SW40, Beckman-Coulter, Inc., Brea, CA, USA) to sediment exosomes. The precipitate was resuspended with PBS and subjected to the same ultracentrifugation conditions to wash the sample. The precipitate was resuspended with PBS, stored at 4 °C for a short period while −80 °C in the long-term [32].

Exosome shapes were observed through transmission electron microscopy (TEM) after purification. Formvar-carbon coated 200-mesh EM copper grids was floated on the freshly isolated exosomes. After 1 min the exosomes were absorbed into the copper membrane. Next, the copper was transferred to the EM solution drop (uranyl acetate) for negative staining and incubated for 1 min. Then, this was examined under a JEOL JEM-1400 electron microscope (Japan).

### 4.4. Exosome Isolation from Transfection Cultured Cells

For exosome isolation, MFF-1 cells grown in 150 mm cell culture dishes were washed thrice with PBS at approximately 80% confluence and cultivated with DMEM containing 10% FBS depleted bovine serum extracellular vehicles (EVs) by overnight ultracentrifugation at 100,000× *g* at 4 °C after transfection with plasmids. After 48 h, the conditioned medium (CM) was collected and first pre-cleared by centrifugation at 300× *g* for 10 min at RT to remove the floating cells. Additionally, all subsequent centrifugation steps were performed at 4 °C. Next, the supernatant was spun at 20,000× *g* for 20 min to remove dead cells and shedding vesicles. Then, to collect exosomes, the supernatant was isolated by ultracentrifugation at 110,000× *g* for 70 min (Ti70, Beckman-Coulter, Inc., Brea, CA, USA) and the supernatant was removed. Furthermore, the precipitate was resuspended in a large volume of PBS and passed through a 0.22 μm pore PES filter (Merck Millipore, Billerica, MA, USA). Then, this supernatant (pre-cleared medium) was next followed by ultracentrifugated at 110,000× *g* for 70 min (SW40, Beckman-Coulter, Inc., Brea, CA, USA) to sediment the exosomes. The precipitate was resuspended with PBS and subjected to the same ultracentrifugation conditions to wash the sample.

### 4.5. Western Blot Analysis

Cells were collected, washed twice with ice-cold PBS, and solubilized in cell lysis buffer for Western blot analysis and IP (Beyotime, ShangHai, China) were then centrifuged at 10,000× g at 4 °C to remove cellular debris. The number of mandarin fish serum-derived exosomes was measured through the total protein concentration. Briefly, isolated exosomes were re-suspended in PBS, mixed with 20 mM Tris-HCL 1% SDS and then sonicated for five minutes, three times, with vortexing in between [33]; protein concentrations of the samples were determined using the Pierce^TM^ Microplate BCA protein assay kit (Thermo Fisher Scientific, Rockford, IL, USA). Briefly, the contents of one albumin standard ampule (BSA) were diluted into several tubes as a protein standard. The unknown samples were diluted with the same buffer. The working buffer was added, and the tubes were incubated at 37 °C for 30 min. The absorbance of all the samples was measured within 10 min. The standard curve was used to determine the protein concentration of each unknown sample.

The purified samples (20 μg) were mixed with 5× loading buffer (250 mM Tris-HCl pH 6.8, 10% SDS, 5% β-mercaptoethanol, 50% glycerinum, 0.5% bromophenol blue), boiled for 10 min, and then subjected to SDS-PAGE for separation. Western blot analysis was performed as described previously [34]. Tsg101, Mx1, HspA8, VDAC1, and GAPDH antibodies were purchased from Abcam (Abcam, Cambridge, UK). GFP-tag antibodies were obtained from Proteintech (Wuhan, China). Anti-ISKNV-VP101L (mAb2D8, unpublished data) was obtained from Dr. Dong CF. The mandarin fish CD63 rabbit polyclonal antibodies (anti-CD63) were prepared for this study. The purified his-tag mandarin fish CD63 recombinant proteins were emulsified with equal volumes of Freund’s complete adjuvant (FCA) for the first injection and Freund’s incomplete adjuvant for the two subsequent injections. The rabbit received three subcutaneous injections at 2-week intervals. Blood samples were taken on the 10th day after the last injection for serum collection.

### 4.6. ISKNV Genome Equivalent (GE) Level Determined by qPCR

An absolute quantitative real-time PCR analysis of the DNA copies of *isknv-mcp* was used to measure the ISKNV genome copies in each sample; the copies of the *β-actin* DNA were also measured by absolute quantitative real-time PCR analysis used as an endogenous reference to normalize the cell number variations in each sample [35]. DNA from the infected cells was extracted using DNeasy blood and tissue kits (Qiagen, Valencia, CA, USA). The ISKNV genome was determined using a LightCycler480 instrument (Roche, Mannheim, Germany). Absolute quantitative real-time PCR was performed using *isknv-mcp* gene primers, forward (*isnknv-mcp* F primer 5′-CAATGTAGCACCCGCACTGACC-3′) and reverse (*isnknv-mcp* R primer 5′-ACCTCACGCTCCTCACTTGTC-3′); *β-actin* served as a reference gene to calibrate the cDNA template, and forward (*β-actin* F primer 5′-GGAGTGATGGTCGGTATG-3′) and reverse (*β-actin* R primer 5′-GAAGGTGTGATGCCAGAT-3′) primers. The PCR reaction mixture (10 μL) contained 5 μL 2 × SYBR Premix Ex Taq (TaKaRa, DaLian, China), 1 μL DNA template, 0.2 μL of 10 μM primers, and 3.6 μL H_2_O. The quantitative real-time PCR conditions were as follows: one cycle at 95 °C for 10 s, 40 cycles of 5 s at 95 °C, 40 s at 60 °C, and 1 s at 72 °C. qPCR was performed at three replicates per sample [35].

### 4.7. Proteomics and Data Analysis

Exosomes isolated from mandarin fish serum were harvested and lysed. After the protein concentration was detected with the BCA assay, peptides obtained after digestion were subjected to LC-MS/MS analysis in the Research Center for Proteome Analysis (Shanghai, China). LC-MS/MS analysis was performed using a Q Exactive mass spectrometer (Thermo Fisher Scientific, Rockford, IL, USA) that was coupled to Easy nLC (Thermo Fisher Scientific, Rockford, IL, USA) for 60/120/240 min (determined by project proposal). The mass spectrometer was operated in positive ion mode. MS data were acquired using a data-dependent top10 method, which dynamically chose the most abundant precursor ions from the survey scan (300–1800 *m*/*z*) for HCD fragmentation. The automatic gain control (AGC) target was set to 3e6, and the maximum inject time to 10 ms. The dynamic exclusion duration was 40.0 s. Survey scans were acquired at a resolution of 70,000 at *m*/*z* 200, the resolution for HCD spectra was set to 17,500 at *m*/*z* 200, and the isolation width was 2 *m*/*z*. Normalized collision energy was 30 eV and the underfill ratio, which specifies the minimum percentage of the target value likely to be reached at the maximum fill time, was defined as 0.1%. The instrument was run with peptide recognition mode enabled. Finally, the proteins were analyzed with triple TOP DDA or SWATH.

### 4.8. Observation of Exosomes in the MFF-1 Cells by Confocal Microscopy

MFF-1 cells were grown in 24-well plates with glass coverslips at density of 1 × 10^5^/one well. Cells were incubated with PKH67 (Sigma, St. Louis, MO, USA ) marked exosomes for 2 h and then incubated with Hoechst 33342 (Thermo Fisher Scientific, Rockford, IL, USA). Images were taken with a confocal microscope (Zeiss LSMS7 DUO NLO, Heidenheim, Germany). Cells were grown at low density on 24-well plates with glass coverslips. Cells were incubated with Mx1-GFP-tag exosomes for 2 h. The media were replaced with exosomes-free media containing 100 nm Lysotracker Red DND-99 (Thermo Fisher Scientific, Rockford, IL, USA). After incubation for 1 h, cells were fixed with 4% paraformaldehyde for 15 min at room temperature (RT). After washing with PBS three times, the coverslips were blocked with 5% normal goat serum solution for 30 min. The coverslips were then washed several times with phosphate-buffered saline with Tween 20 and incubated with Hoechst 33342 (Invitrogen, Waltham, MA, USA). Images were taken with a confocal microscope (Zeiss LSMS7 DUO NLO, Germany). Mock-incubated cells were similarly stained as controls.

## Figures and Tables

**Figure 1 ijms-22-10346-f001:**
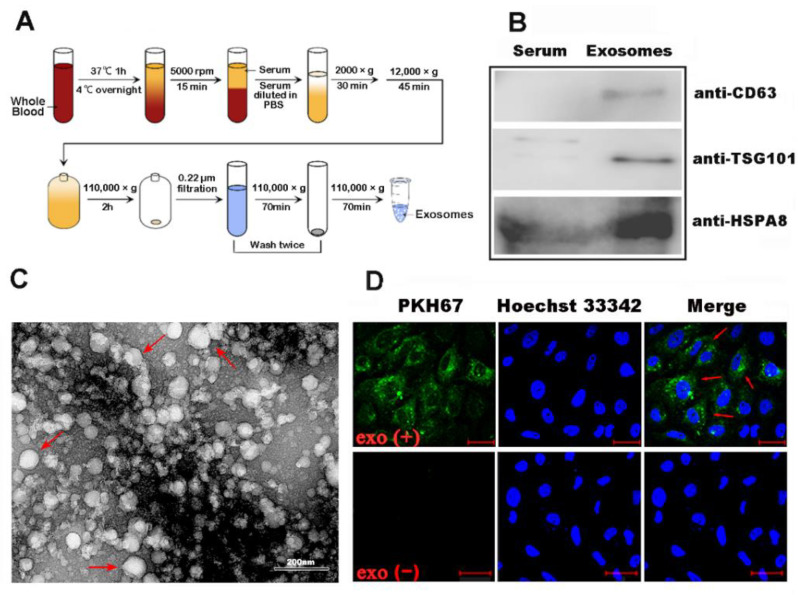
Isolated exosomes from mandarin fish serum. (**A**) Schematic of the generation of purified serum-derived exosomes from mandarin fish by using differential centrifugation. (**B**) Purified exosomes (right) from mandarin fish serum were analyzed using Western blot with antibody directed against CD63, TSG101, and HspA8. The normal untreated mandarin fish serum (left) was used as the control. (**C**) Transmission electron microscopy observations of the purified exosomes from serum. (**D**) Confocal microscopy assay showed that the exosomes were internalized by MFF-1 cells after incubation with PKH67-labeled exosomes for 2 h.

**Figure 2 ijms-22-10346-f002:**
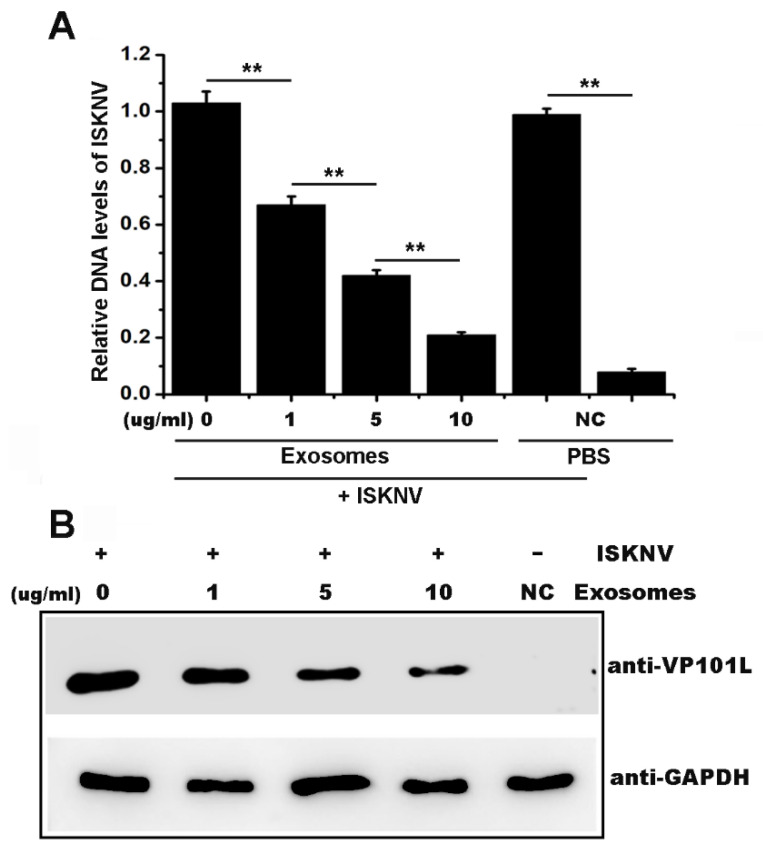
Mandarin fish serum-derived exosomes inhibit ISKNV infection. (**A**) Cells were incubated with 0, 1, 5, and 10 μg/mL exosomes after infection with ISKNV at a MOI of 1 (lanes 1–4); incubation with PBS as positive control (lane 5) and uninfected cells (lane 6) as negative control (NC); the DNA copies of *isknv-mcp* were quantified via absolute quantitative real-time PCR in MFF-1 cells at 48 h post infection. The copies of *β-actin* DNA were also measured by absolute quantitative real-time PCR analysis used as an endogenous reference to normalize the cell number variations in each sample. The y-axis represents the DNA copies of *isknv-mcp*/the DNA copies of *β-actin*. Vertical bars represent statistical significance as indicated by asterisks, with ** referring to *p* < 0.01. (**B**) Western blot analysis of the ISKNV-VP101L protein after 48 h of infection.

**Figure 3 ijms-22-10346-f003:**
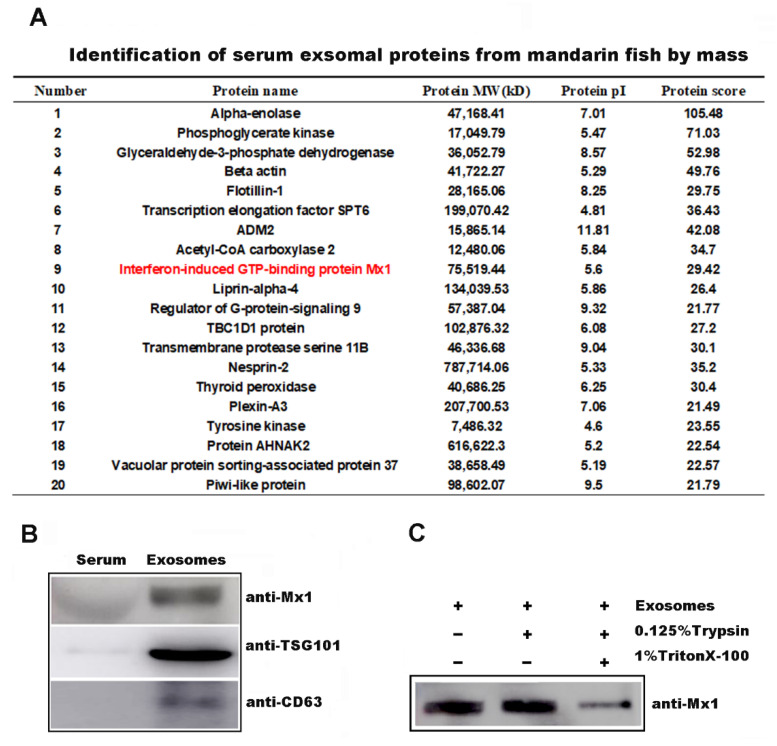
The protein composition of mandarin fish serum-derived exosomes. (**A**) Purified serum-derived exosomes from mandarin fish were analyzed with LC-MS/MS to determine the host proteins. (**B**) Mx1 proteins presented in serum-derived exosomes were confirmed by Western blots with antibody directed against Mx1. Left: untreated mandarin fish serum. Right: purified serum-derived exosomes. (**C**) The assay of exosome resistance to trypsin digestion. The same amounts of exosomes were used for each condition (10 μg); line 1 is control, mock treated (exo); line 2 is incubated with trypsin (0.125% Trypsin) for 30 min at room temperature; line 3 is treated with 1% Triton X-100 for 5 min before incubation with trypsin (0.125% trypsin). Western blot analysis with antibody directed against Mx1.

**Figure 4 ijms-22-10346-f004:**
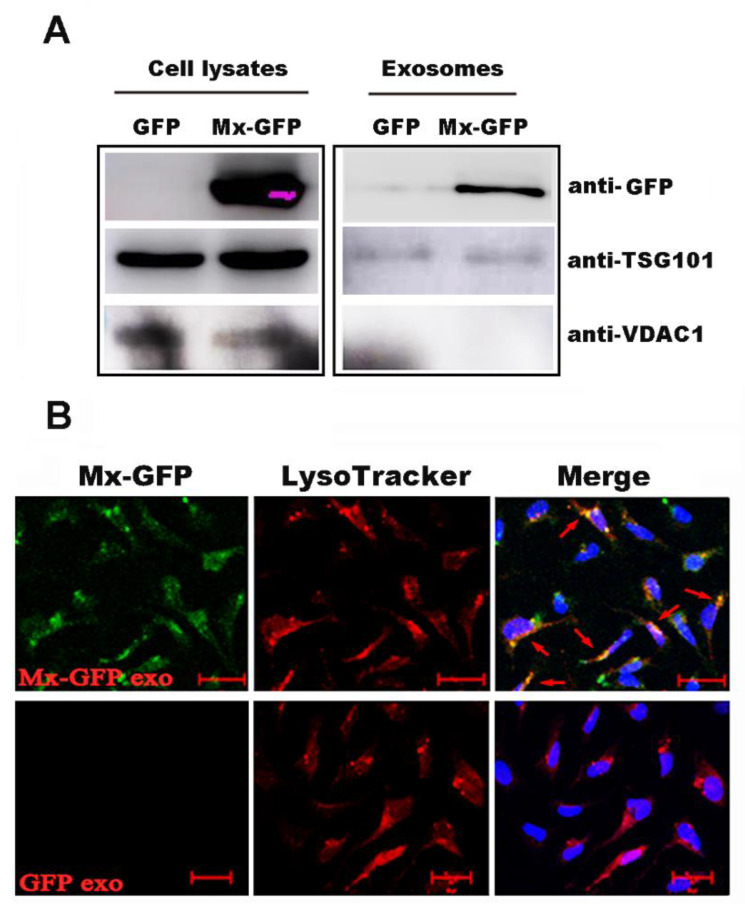
Mx1 protein transferred to the recipient cells though the exosomes. (**A**) Cells were transfected with Mx1-GFP-expressing vectors secreted exosomes with Mx1. Exosomes were isolated from cell cultures from transiently transfected MFF-1 cells. Cellular and exosomal lysates were analyzed by Western blot for the presence of Mx1, TSG101, and voltage-dependent anion channel protein 1 (VDAC1, a mitochondrial marker). (**B**) Confocal microscopy showed the Mx1-GFP exosomes were internalized by MFF-1 cells. The cells were transfected with an Mx1-GFP-expressing vector, and the exosomes were isolated from culture supernatant. Exosomes were incubated with MFF-1 cells for 2 h and then with LysoTracker (LT) for 1 h.

## Data Availability

Data available in a publicly accessible repository.

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
