# Peer review of "Evidence for a Novel Antiviral Mechanism of Teleost Fish: Serum-Derived Exosomes Inhibit Virus Replication through Incorporating Mx1 Protein"

_ijms, 2021, doi:10.3390/ijms221910346_

Round 1
Reviewer 1 Report
Prevention and treatment of viral infections are urgently needed. However, high mutation rate of viruses and multiple drug resistance exceed limited repertoire of currently available antivirals. Therefore, design and development of new approaches by using bionanotechnology methods are topical.
Because of absence of important details in Introduction the manuscript design remains uncertain. Thus, taxonomy status of Infectious spleen and kidney necrosis virus (ISKNV) as well as molecular organization of the double-stranded DNA containing iridovirus was not described. Moreover, mechanism of antiviral action of MX1 also remains unclear despite available scientific data. Innate immunity refers to nonspecific defense mechanisms induced within a period from a few minutes until several hours of an antigen's appearance, processing and presentation. MX1 belongs to unspecific resistance but not adaptive immune response. Antiviral properties of MX1 were shown for several mainly RNA-containing viruses but not against iridoviruses.
Specific comments.
- Methods
Section 4.1. Cells, virus, and animals should be described in details.
Origin of cells and ISKNV, corresponding GenBank accession numbers and references. How virus species was determined? How many laboratory passages? What about laboratory host?
Sample preparation for TEM should be described more precisely. Why TEM images of the virus ISKNV were not shown?
How multiplicity of infection (MOI) was determined? How concentrations of exosomes were measured? It seems important to compare amounts of virions and numbers of exosomes.
4.3. Western blot analysis
How ISKNV-ORF101 and sc-CD63 antibodies were prepared?
Results
Fig. 1, C
It is hardly possible to estimate size range of exosomes. Histogram might show real diameters.
Fig. 1, D
Fluorescent confocal microscopy revealed poor penetration of labelled exosomes in part of cells but not in all of them. Probably, concentration of exosomes was not enough? How many exosomes per each cell were added?
Antiviral properties were analyzed by qPCR and Western blotting.
"As shown in Figure 1E, the DNA level of isknv-mcp obviously decreased after incubation with exo-somes in a dose-dependent manner. Similar results were also observed in the levels of the ISKNV VP101L protein (Figure 1F)"
But Figure 1 consists of ABCD. Neither part E, nor F were found there.
Quantitation and normalization of qPCR data are not described. It is especially important for PCR with intercalated unspecific dye SYBR instead of specific TaqMan probes.
Map of the recombinant plasmid Mx1-GFP-tag was not shown.
"The results showed that the Mx1-GFP recombinant protein was detected in the cellular exosomes (Figure 1J), indicating that this protein was packaged into the exosomes."
But the Figure 1J is missing.
Moreover, origin of cloned gene and MX1 recombinant protein (fish or mammalian or something else) remains unclear.
Therefore, both discussion and conclusion do not correspond to currently demonstrated results with uncertain methods.
Author Response
Dear reviewer,
Thank you for review our submitted paper comments, ijms-1308187 ["An evidence for a novel antiviral mechanism of teleost fish: serum-derived exosomes inhibit virus replication through incorporating Mx1 protein"]. The suggestions from you were very helpful in the revision of our manuscript. Based on the comments and suggestions, we now revised the manuscript. The response to your comments in the attachment PDF.
Thank you very much for your review this manuscript.
Sincerely yours,
Chang-jun Guo

Reviewer 2 Report
My comments on the manuscript untitled ‘An evidence for a novel antiviral mechanism of teleost fish: serum-derived exosomes inhibit virus replication through incorporating Mx1 protein’ are as follows:
- The manuscript is written very poorly in English, some passages are completely incomprehensible.
- The purpose of the research undertaken was not clearly stated.
- The order of the sections in Materials and Methods is incorrect. It is also difficult to follow the scheme / course of conduct during the studies described in the manuscript.
- The research methodology is described enigmatically. Lack of details in the description of individual procedures. The culture conditions of MFF-1 cells are not known, information on how antibodies against ISKNV-ORF101 and sc-CD63 were prepared (p. 8), no reference to the Western blot procedure (p. 8), no details on the LC-ALDI MS/MS. However sometimes, some details appear unexpectedlyin Results. I do not understand the description in paragraph 4.4 at all. What is 'confocal microscopy assay'?
- The written description of the results is completely different from the graphic documentation. When describing the results, the Authors refer to Figures that are not present in the manuscript (Fig. 1F, G, H, K). Instead, there are Figures 2 and 3 that are not commented on. Poor blot quality in Fig. 1B.
- The Discussion chapter can hardly be considered as a real discussion of the obtained results or their interpretation against the background of the available literature.
- The consent number for the use of fish in experimental work was not included in this manuscript.
- Unexpanded abbreviations appeared as keywords.
- No dots are put at the end of the titles.
Overall, the work is written very poorly in English, and in a chaotic manner. I have the impression that the authors did not bother to carefully read the manuscript prepared for sending to Int. J. Mol. Sci. Otherwise, the errors I mentioned would be caught.
Author Response
Dear reviewer,
Thank you for review our submitted paper, ijms-1308187 ["An evidence for a novel antiviral mechanism of teleost fish: serum-derived exosomes inhibit virus replication through incorporating Mx1 protein"]. The suggestions from you were very helpful in the revision of our manuscript. Based on the comments and suggestions, we now revised the manuscript. The response to your comments in the attachment PDF.
Thank you very much for your review this manuscript.
Sincerely yours,
Chang-jun Guo

Round 2
Reviewer 1 Report
English language extensive editing is still required. My suggestions are in attached file.
Measurements of protein concentrations are described in the revised section "Methods". How exosome concentrations were determined? They are not the same.
Normalization of qPCR data by using beta-actin gene remains under question. How many genes of actin(s) are in fish cells? Why fluorescent hydrolysis probe (TaqMan or something like that) was not used instead of SGI intercalating dye?
In order to calculate multiplicity of infection (MOI) cell numbers were calculated before seeding? But cell could divide and their amounts significantly grew in several days?
Something is wrong with fluorescent dye to label cellular nuclei for confocal fluorescent microscopy. Two different dye are described in methods and results.

Author Response

(The authors gave the same response as above.)

Reviewer 2 Report
My comments to the revised version of the manuscript untitled ‘An evidence for a novel antiviral mechanism of teleost fish: serum-derived exosomes inhibit virus replication through incorporating Mx1 protein’ are as follows:
- Contrary to the authors' assurances (point 9), I can still see that the title of the article ends with a comma (line 4) .
- English language and style - I don't see any significant improvement. Extensive editing of English language and style is still required.
For example lines 17-22:
Is:
Exosomes were isolated from mandarin fish serum by ultra-centrifugation could internalize into Mandarin fish fry (MFF-1) cells and inhibited Infectious spleen and kidney necrosis virus (ISKNV) infection. To further investigated the underlying mechanisms of exosomes in inhibiting ISKNV infection. The protein composition of serum-derived exosomes was by analysis mass spectrometry and found that myxovirus resistance 1 (Mx1) was incorporated in the exosomes. Furthermore, the scMx1mandarin fish Mx1 protein was proved transferred to the recipient cells though the exosomes.
Suggested changes:
Exosomes were isolated from mandarin fish serum by ultra-centrifugation could internalize into were internalized by Mandarin fish fry (MFF-1) cells and inhibited were able to inhibit Infectious infectious spleen and kidney necrosis virus (ISKNV) infection. To further investigated the underlying mechanisms of exosomes in inhibiting ISKNV infection (this sentence is incomplete). The protein composition of serum-derived exosomes was by analysis analyzed by mass spectrometry and . It was found that myxovirus resistance 1 (Mx1) was incorporated in by the these exosomes (or simply: exosomes). Furthermore, the scMx1mandarin fish Mx1 protein was proved to be transferred to into the recipient cells though the these exosomes (or simply: exosomes).
Author Response

(The authors gave the same response as above.)
